# Anatomical Proposal for Botulinum Neurotoxin Injection for Glabellar Frown Lines

**DOI:** 10.3390/toxins14040268

**Published:** 2022-04-10

**Authors:** Kyu-Ho Yi, Ji-Hyun Lee, Hye-Won Hu, Hee-Jin Kim

**Affiliations:** 1COVID-19 Division, Wonju Public Health Center, Wonjusi 26493, Korea; kyuho90@korea.kr; 2Division in Anatomy and Developmental Biology, Department of Oral Biology, Human Identification Research Institute, BK21 PLUS Project, College of Dentistry, Yonsei University, 50-1 Yonsei-ro, Seoul 03722, Korea; jh_anatomy@yuhs.ac (J.-H.L.); wonhuh@yuhs.ac (H.-W.H.); 3Department of Materials Science and Engineering, College of Engineering, Yonsei University, Seoul 03722, Korea

**Keywords:** corrugator supercilii muscle, botulinum neurotoxin, glabellar frown line, facial wrinkle, injection point, ptosis, samurai eyebrow

## Abstract

Botulinum neurotoxin injection for treating glabellar frown lines is a commonly used method; however, side effects, such as ptosis and samurai eyebrow, have been reported due to a lack of comprehensive anatomical knowledge. The anatomical factors important for the injection of the botulinum neurotoxin into the corrugator supercilii muscle has been reviewed in this study. Current understanding on the localization of the botulinum neurotoxin injection point from newer anatomy examination was evaluated. We observed that for the glabellar-frown-line-related muscles, the injection point could be more accurately demarcated. We propose the injection method and the best possible injection sites for the corrugator supercilii muscle. We propose the optimal injection sites using external anatomical landmarks for the frequently injected muscles of the face to accelerate effective glabellar frown line removal. Moreover, these instructions would support a more accurate procedure without adverse events.

## 1. Introduction

The botulinum neurotoxin (BoNT) mechanism of action works by binding presynaptically to a high-affinity protein receptor and polysialogangliosides on the cholinergic nerve terminals, then entering the neuron and causing a sustained inhibition of synaptic transmission [1,2]. Clinically, BoNT is aesthetically used for removing facial lines, as it relaxes the muscles used in facial expression. However, the glabella frown line is of more concern to many individuals (Figure 1).

Glabellar frown lines are created by three muscles: the corrugator supercilii (CSM), procerus and depressor supercilii muscles; however, the most involved muscle is the CSM. The superomedial fibers of the orbicularis oculi, otherwise known as the depressor supercilii, are interconnected with the CSM and procerus muscles (Figure 2).

Side effects of BoNT injection in facial muscles, such as ptosis and samurai eyebrows, have been reported to be due to lack of subtle anatomical information. Although susceptibility to BoNT varies among individuals, there is no efficient cure for ptosis, which lasts for several months. Consequently, a safer method is to begin the primary treatment at a lower dosage. If the desired outcomes are not achieved, an additional touchup treatment could follow.

Subsequently, a modified dosage may be applied for each treatment. While injecting BoNT into the CSM, anatomical structures should be meticulously understood to prevent side effects, such as undesirable palsy of adjoining muscles, which may result in ptosis and samurai eyebrow [3,4,5]. Ptosis may occur as the BoNT diffuses into the levator palpebrae superioris muscle, following injection into the CSM. Consequently, the dose must be adjusted based on the muscle area of the person to prevent the severe side effects mentioned above.

Furthermore, increased doses and repeated injections of BoNT produce antibodies, leading to unsatisfactory treatment outcomes [6,7,8,9]. Various studies relating BoNT injection points to specific muscle anatomy have already been published [10].

This study aimed to suggest safe and effective BoNT injection points and injective methods for the CSM of glabellar frown lines.

## 2. Anatomy of the CSM

The CSM, originating in the superomedial part of the bony orbital rim, passes superolaterally at an angle of 30°, and runs to the middle of the eyebrow, attaching to the dermal layer of the skin [11]. The CSM lies under the frontalis muscle as it originates from the frontal bone.

Specifically, the CSM originates from about 16 mm above the horizontal intercanthal plane (HL) and 4–14 mm from the midline, inserts into the dermis (30 mm above the HL and 16–35 mm from the midline), and interlinks into the frontalis muscle (Figure 3) [11].

The vertical length of the CSM in Asians (15 mm) is shorter than that in Caucasians (21 mm) [11]. The CSM consists of two muscle bellies: oblique and transverse bellies [11]. The oblique belly runs vertically, whereas the transverse belly runs horizontally. However, there are two types of oblique bellies: narrow and broad (Figure 4).

The oblique belly of the narrow type extends in a narrow rectangular shape [11]. This type of belly merges with the frontalis muscle within the medial third of the transverse belly. The broad type is triangular, covering most of the transverse belly superficially. The broad oblique belly tends to develop more powerful muscle contractions of the eyebrow above the mid pupillary line. However, two types of the CSM muscle are difficult to distinguish and the injection points for two types are exactly the same.

The CSM pulls the eyebrows downward and medially and produces vertical lines while frowning [12]. If the CSM is paralyzed, effacement of the glabellar lines and widening of the eyebrows are observed.

## 3. Proposed Injection Techniques

The glabellar muscles are not apart from each other; instead, these muscles are interconnected [11]. Consequently, the BoNT injected into the CSM may diffuse to the nearby muscles. It is proposed that an injection be administered at four points: medial and lateral points [13].

BoNT was injected into the medial and lateral points of the CSM, as shown in a standard form; for the medial injection, the needle was advanced deeply until it touched the periosteum, withdrawn 2–3 mm, followed by slow injection of BoNT into the muscle [13]. A Subdermal injection is preferable for the lateral points, and it is recommended to press the frontal notch with fingers while injecting to avoid the inflow of BoNT into the orbital cavity [14]. Additionally, the injection needle should point upward.

A total of 6 U is injected; 1.5 U into each medial and lateral point [14]. The medial point is an interciliary point located on the frontal notch that can be palpated. The lateral point is the crossing point of the mid-pupillary line and superciliary arch (Figure 5).

## 4. Side Effects

### 4.1. Ptosis

Treatment of glabellar frown lines with BoNT in 264 subjects included adverse effects, such as transient headache (15%) and mild unilateral blepharoptosis (5.4%), resolved mostly in one month [15]. In a meta-analysis by Brin et al., adverse effects, including eyelid ptosis (1.8%) and eyelid edema were observed with BoNT, for the treatment of glabellar wrinkles [16].

BoNT injected into the glabellar area can diffuse downward, paralyzing the levator palpebrae superioris muscle, which causes ptosis and is usually unilateral (Figure 6).

Almost 30–40% of the cornea is covered in the upper eyelid in patients with ptosis. Early literature described a 1–3% prevalence of ptosis, while an approved FDA clinical study reported a prevalence of 5.4% [15,16]. This occurs when the lateral injection point is too deep or when too much BoNT is injected. Therefore, lateral injections should be performed superficially in the subdermal layer. When ptosis occurs, patients should be reassured that they will regain their eye-opening function. The eyelid should be elevated and ptosis should disappear completely in mild cases. Ptosis can occur for up to 2–3 months; thus, careful initial treatment is required.

For symptomatic treatment, eye drops, including an α-adrenergic agonist that activates the Müller muscle, should be administered three times daily [13]. The Müller muscle, situated between the levator palpebrae superioris and mucosa, has the auxiliary function of eyelid lifting. As the Müller muscle is innervated by sympathetic neurons, eye drops containing 0.5% apraclonidine, an α-adrenergic agonist, can contract the Müller muscle after it has been absorbed into the mucosa and can elevate the eyelid.

### 4.2. Samurai Eyebrow

Samurai eyebrows are adverse effects that result from the diffusion of BoNT affecting the medial part of the frontalis muscle (Figure 7).

Such diffused injections of BoNT paralyze the medial part of the frontalis muscle belly, resulting in the lateral eyebrows moving upward and a subsequent angry appearance [12]. Lee et al. suggested that when samurai eyebrows occur, additional BoNT should be injected along the superior temporal line, based on the established injection method. [17]

## 5. Discussion

To prevent adverse effects, clinicians should pay attention to various factors, such as the injection point, dose of BoNT, direction of the needle, and position of the hand. The site of injection is of great importance for the prevention of ptosis, and it should be as superior as possible to the edge of the orbital rim. The location at which the CSM originates is right above the upper inner boundary of the orbital rim. One unit of BoNT is reported to spread over 1.5 to 3 cm; therefore, injecting a bit above the location where the CSM originates is allowable [18,19].

In a study by Yang et al., the origin and insertion points of CSM were anatomically studied [11]. The CSM originates 16 mm above the horizontal intercanthal plane and 4–14 mm from the midline, inserts into the dermis (30 mm above the HL and 16–35 mm from the midline) and interlinks with the frontalis muscle.

In patients with chronic migraine, the CSM is targeted for BoNT injection because it is related to the entrapment of the supratrochlear nerve within the CSM. Clinically, 5 U of BoNT injection per CSM has been recommended, according to the PREEMPT trial [20]. In the study, Lee et. al. proposed that injecting a lower dose of 5 U at a defined anatomical site produces the same therapeutic effect with a safer side effect profile [21].

In the clinical field, many injections are performed on the eyebrows, as they are usually located along the upper inner boundary of the orbital rim [14]. However, in elderly patients, if the location of injection is selected based on the eyebrows, there is a high probability of ptosis, as their eyebrows are lowered during the aging process [13]. In patients with lowered eyebrows, one can mistakenly inject below the boundary of the orbital rim. The injection was performed slowly to avoid the BoNT from dispersing down to the eyelids [13]. During the injection, manual blocking was performed along the inner boundary of the orbital rim [13].

Broad and precise anatomical knowledge of the muscles is essential for attaining the maximum effect with the lowest possible amount of BoNT. We reviewed the strategies used in previous studies, and the injective strategy varied in each study. As wrinkle correction via BoNT is frequently conducted, side effects, such as paralysis of adjacent muscles and ptosis, have been reported.

The limitation of this study is that the review was based on anatomical information and was not a clinical study. Further, clinical study should be conducted. However, the study is significant in the usage of a minimal amount of BoNT and the location is based on anatomical information that can be easily applied.

## 6. Conclusions

In summary, 6 U of CSM BoNT should be injected, with 1.5 U injected into each medial and lateral point.

The medial point is the interciliary point located on the frontal notch that can be palpated. The lateral point is the crossing point of the mid-pupillary line and superciliary arch (Figure 5).

This study performed an extensive analysis of published research on the anatomy of the CSM to provide an anatomical proposal for glabellar frown line correction.

## Figures and Tables

**Figure 1 toxins-14-00268-f001:**
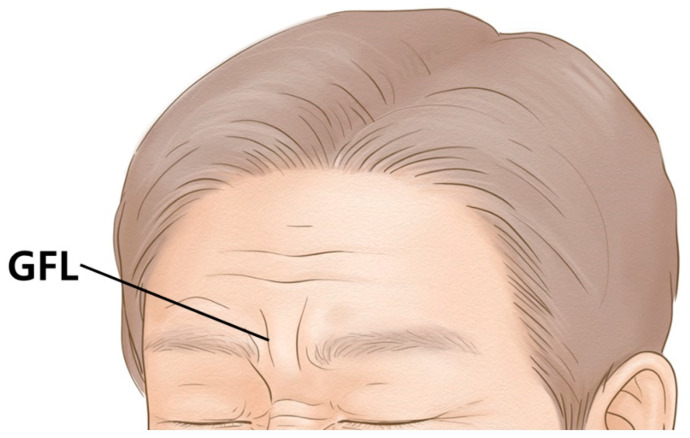
Schematic image of the glabellar frown lines (GFL).

**Figure 2 toxins-14-00268-f002:**
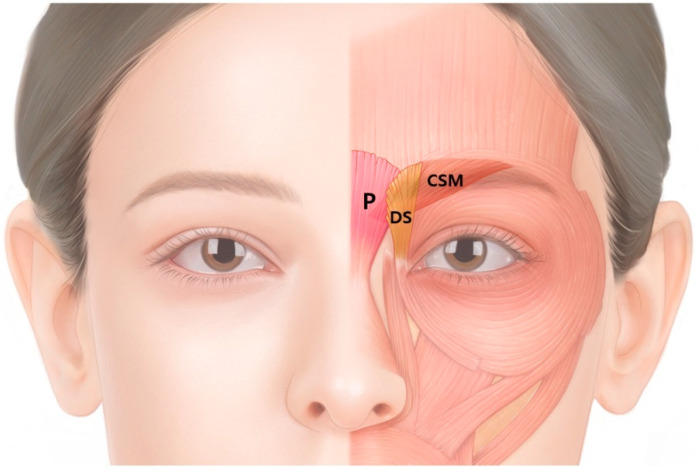
Schematic image of the corrugator supercilii (CSM), procerus (P) and depressor supercilii (DS) muscles. Glabellar frown lines are produced by three muscles; however, the most involved muscle is the CSM.

**Figure 3 toxins-14-00268-f003:**
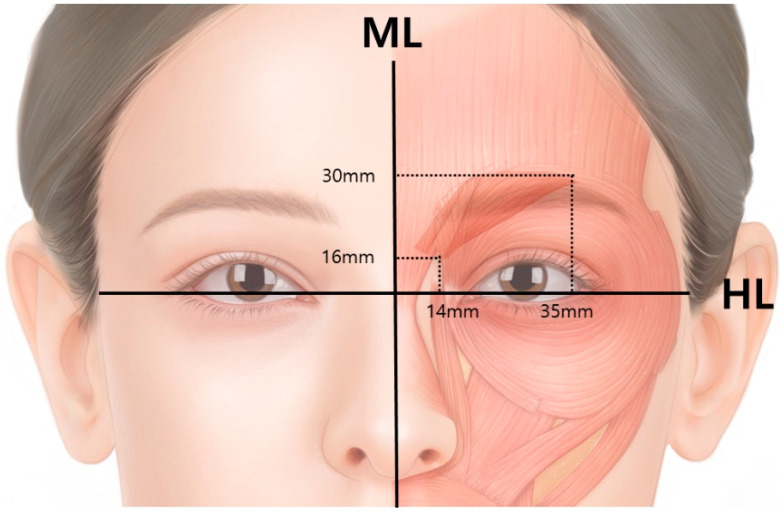
The corrugator supercilii muscle originates from 16 mm above the horizontal intercanthal plane (HL) and 4–14 mm from the midline (ML), inserts into the dermis (30 mm above the HL and 16–35 mm from the midline), and interlinks with the frontalis muscle.

**Figure 4 toxins-14-00268-f004:**
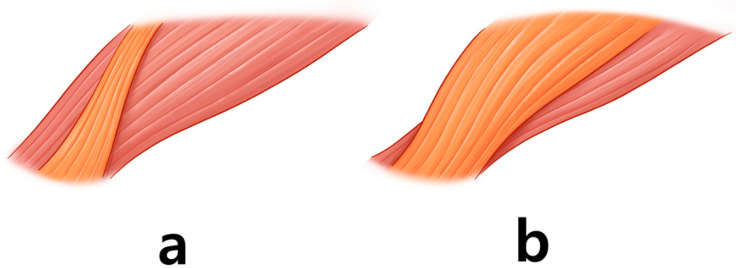
The corrugator supercilii muscle has an oblique belly (orange-colored) and a transverse belly (red-colored). The oblique belly has two types: the narrow type (**a**) and the broad type (**b**).

**Figure 5 toxins-14-00268-f005:**
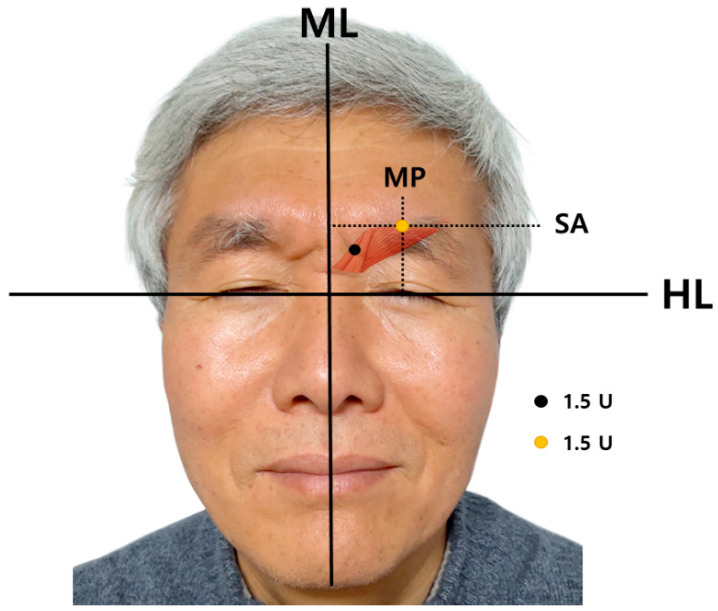
Guidance for injecting botulinum neurotoxin into the corrugator supercilii muscle. A total of 6 U is injected; each medial and lateral points is injected with 1.5 U. The medial point (black dot) is interciliary point located on frontal notch that can be palpated. The lateral point (orange dot) is the crossing point of mid-pupillary line and superciliary arch. Consent was obtained by the volunteer. (ML; midline, HL; horizontal intercanthal line, MP; mid-pupillary line, SA; superciliary arch).

**Figure 6 toxins-14-00268-f006:**
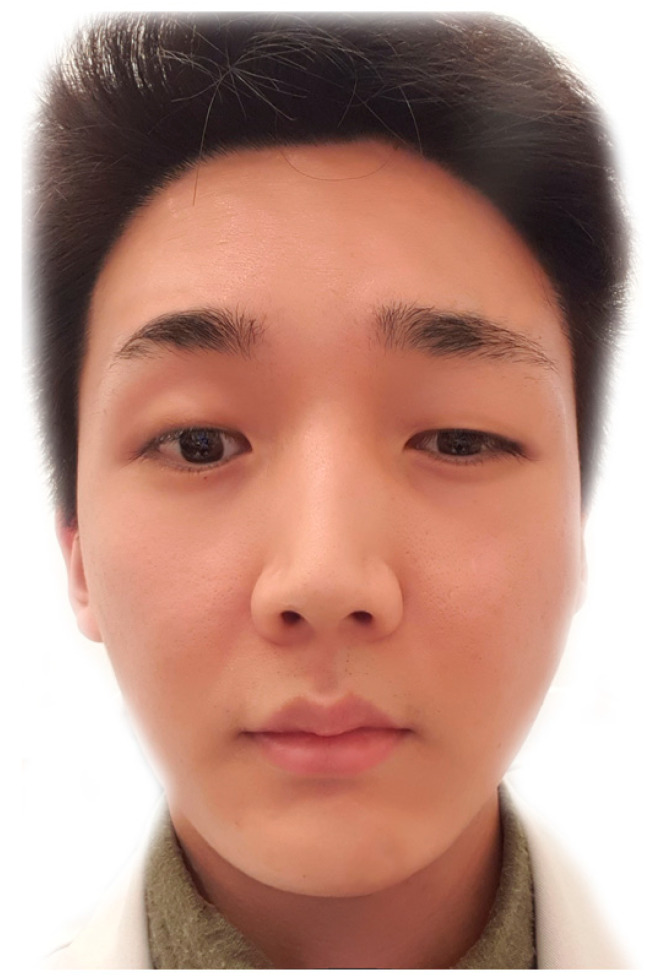
A person with ptosis after botulinum neurotoxin injection into the corrugator supercilii muscle. Consent was obtained from the volunteer.

**Figure 7 toxins-14-00268-f007:**
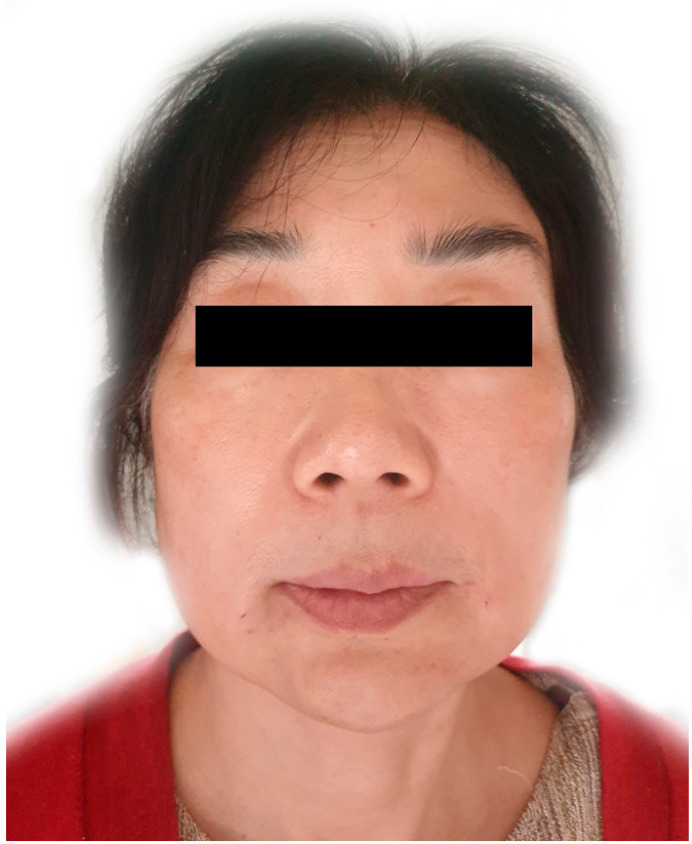
A person with samurai eyebrows after botulinum neurotoxin injection into the corrugator supercilii muscle. Consent was obtained from the volunteer.

## Data Availability

Not applicable.

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
