# Peer review of "Anatomical Proposal for Botulinum Neurotoxin Injection for Glabellar Frown Lines"

_toxins, 2022, doi:10.3390/toxins14040268_

Round 1
Reviewer 1 Report
Overall, please check the entire article and correct spellings for “supercili,” “supercilia,” “supercilii” as appropriate.
Introduction:
- In Line 24-25, “by stimulating the release of acetylcholine…” incorrectly described the mechanism of action of BoNT.
- In Line 35, consider changing to “Side effects of BoNT injection in facial muscles, such as ptosis and samurai eyebrows, have been….”
- In Line 45, it wasn’t clear to me what “…while injection into the CSM; samurai eyebrow, frontalis muscle” means. Is it a punctuation error or an incomplete sentence?
- In Line 78-79, “The CSM pulls the eyebrows downward and medially and contributes vertical while frowning” is similar to “The movement of the CSM shifts the eyebrow medially and inferiorly, making perpendicular lines on the forehead as seen when grimacing” in Line 56-57.
Injection techniques:
- In Line 91, should it be “2. Injection Techniques”?
- Some of the information in Line 109-114 was a duplicate with that in Line 95-98.
- Incomplete sentence in Line 116.
- Please add references for “BoNT spreads approximately 1 cm” in Line 155.
- Only injection techniques for CSM were mentioned. No injection techniques were reported for the procerus and the depressor supercilia muscles. Based on Line 51-52, it appeared to be intended. If so, the article needs to specify in the title that it only focused on injection techniques for the CSM.
- Before “Ptosis,” should there be a subsection title?
- It wasn’t clear to me whether the proposed technique was based on previous studies or the authors’ original idea as there were no citations associated with the “Injection Techniques”. If the former, please add references; if the latter, would be better to clearly state “we propose.”
Discussion:
- In the discussion, the author(s) repeated much information mentioned previously. The author(s) were not able to demonstrate why the techniques they proposed were superior and safer to the others as “various studies relating BoNT injection points to specific muscle anatomy have already been published (Line 49-50).”
- Statements regarding the above problems and limitations of the review need to be honestly mentioned.
Figures:
Very nicely drawn.
Figure 3 legend, “orange-colored.” (Line 71).
Author Response
The authors are very grateful to the reviewer for your meticulous reading of the manuscript. The authors would like to kindly acknowledge your comments.
Review 1
Overall, please check the entire article and correct spellings for “supercili,” “supercilia,” “supercilii” as appropriate.
Answer: We have corrected the spelling as “supercilia”, which is anatomically correct term. Thank you for noticing our mistake.
Introduction:
- In Line 24-25, “by stimulating the release of acetylcholine…” incorrectly described the mechanism of action of BoNT.
- In Line 35, consider changing to “Side effects of BoNT injection in facial muscles, such as ptosis and samurai eyebrows, have been….”
- In Line 45, it wasn’t clear to me what “…while injection into the CSM; samurai eyebrow, frontalis muscle” means. Is it a punctuation error or an incomplete sentence?
- In Line 78-79, “The CSM pulls the eyebrows downward and medially and contributes vertical while frowning” is similar to “The movement of the CSM shifts the eyebrow medially and inferiorly, making perpendicular lines on the forehead as seen when grimacing” in Line 56-57.
Answer:
In Line 24-25, we have described the mechanism of action of BoNT as follows “The Botulinum neurotoxin (BoNT) mechanism of action is the permanent blockage of acetylcholine release”.
In line 35, as of your suggestion, the phrase has been changed as “Side effects of BoNT injection in facial muscles, such as ptosis and samurai eyebrows, have been reported to be due to lack of subtle anatomical information”.
In the line 45, the phrase is corrected as following “Ptosis may occur as the BoNT diffuses into the levator palpebrae superioris muscle, while injection into the CSM.”.
In the line 56-57, “The movement of the CSM shifts the eyebrow medially and inferiorly, making perpendicular lines on the forehead as seen when grimacing” has been deleted due to repeated description.
Injection techniques:
- In Line 91, should it be “2. Injection Techniques”?
- Some of the information in Line 109-114 was a duplicate with that in Line 95-98.
- Incomplete sentence in Line 116.
- Please add references for “BoNT spreads approximately 1 cm” in Line 155.
- Only injection techniques for CSM were mentioned. No injection techniques were reported for the procerus and the depressor supercilia muscles. Based on Line 51-52, it appeared to be intended. If so, the article needs to specify in the title that it only focused on injection techniques for the CSM.
- Before “Ptosis,” should there be a subsection title?
- It wasn’t clear to me whether the proposed technique was based on previous studies or the authors’ original idea as there were no citations associated with the “Injection Techniques”. If the former, please add references; if the latter, would be better to clearly state “we propose.”
Answer:
In Line 91, we have changed as your point. “2. Injection Techniques” is correct.
In Lines of 95-98 has been deleted due to duplicated information. As well the line 109-114 has been relocated.
In the Line 116, the incomplete sentence has been revised “Treatment of glabellar frown lines with …”.
In Line 155, we have cited reference for Line 155, “Diffusion, spread, and migration of botulinum toxin”.
We definitely agree with your point of view, thereby, we have changed the sub-title and focused on the corrugator supercilia muscle only. We have deleted the parts for procerus and depressor supercilia, for these muscle are not major concern for the glabellar frown line. However, other reviewer has requested the figure of these muscles, thereby we have additional figure (Figure 2) in the revised version of the manuscript.
We have added “Side-effect” subtitle as reviewer has suggested. We have changed the injection techinique to “Proposed injection technique”. As well, we have changed the title to “Anatomical proposal for botulinum neurotoxin injection for glabellar frown lines”.
Discussion:
- In the discussion, the author(s) repeated much information mentioned previously. The author(s) were not able to demonstrate why the techniques they proposed were superior and safer to the others as “various studies relating BoNT injection points to specific muscle anatomy have already been published (Line 49-50).”
- Statements regarding the above problems and limitations of the review need to be honestly mentioned.
Answer: We have deleted the information previously mentioned as the reviewer requested. We have added a paragraph of “limitation”. The authors stated “The limitation of the study was the review was based on the anatomical information and was not clinical study. Further, clinical study should be conducted. However, the study is significant in usage of minimal amount of BoNT and location are based on anatomical information that can be easily applied.”
Figures:
Very nicely drawn.
Figure 3 legend, “orange-colored.” (Line 71).
Answer: We have changed the Figure 3 legend with “orange-colored”. As well, additional figure has been added. Actual patient cases has been replaced for the Figure 5 and 6.
We are grateful to have your helpful comments and for the precious time spent reviewing our article. According to your suggestions, we have deleted parts of other minor muscles (procerus and depressor supercilia) and deleted word “guideline” that may mislead the practitioner. Rather we have used proposal based on the anatomical information. Additionally, we have changed figures with real patient photography. We are confident that readers would be interested in this revised article.
Reviewer 2 Report
This paper reads well. It provides useful guideline for BoNT treatment of GFL. I have a few suggestions.
1) Although it is helpful to know the distance/anatomical landmark in absolute measures (in cm), it is difficult to apply this knowledge clinically, since the size/distance is different for individual patients. I would recommend that the authors provide guidelines based on anatomical landmarks, something like this, for example, on Figure 2, key landmarks could be defined first, the intersection between ML and HL, and the medial 1/3 of eyebrow. The injection site is then defined as the midway of the line that connects these two points.
2) Anatomy of all involved muscles (CSM, procerus and depressor supercili) in one picture would be helpful to understand the important of CSM and how complications might happen.
3) pictures of real cases (successful cases and cases with complications - Fig 5&6) would be better that schematic images if possible.
Author Response
The authors are very grateful to the reviewer for their meticulous reading of the manuscript. The authors would like to kindly acknowledge the review. Here follow our answers to specific points from the reviewers.
This paper reads well. It provides useful guideline for BoNT treatment of GFL. I have a few suggestions.
- Although it is helpful to know the distance/anatomical landmark in absolute measures (in cm), it is difficult to apply this knowledge clinically, since the size/distance is different for individual patients. I would recommend that the authors provide guidelines based on anatomical landmarks, something like this, for example, on Figure 2, key landmarks could be defined first, the intersection between ML and HL, and the medial 1/3 of eyebrow. The injection site is then defined as the midway of the line that connects these two points.
Answer: As the reviewer suggestion we have set up anatomical landmarks that can be easily noticeable for the practitioner. The medial point should be interciliary point located on infraorbital notch that can be palpated. The lateral point was crossing point of mid-pupillary line and superciliary arch as the suggestion. Thank you for suggesting great idea!
- Anatomy of all involved muscles (CSM, procerus and depressor supercili) in one picture would be helpful to understand the important of CSM and how complications might happen.
Answer: The figure 2 has been added as of the request. Since other reviewers have suggested focusing on the CSM muscle (main muscle), other muscles have been deleted, however, these muscles still minorly involves in glabellar frown line, we have additional figure (Figure 2) that involves procerus and depressor supercilii muscle.
- pictures of real cases (successful cases and cases with complications - Fig 5&6) would be better that schematic images if possible.
Answer: We have changed the images with the patient photography of complications (ptosis and samurai eyebrow) for the better understanding for the readers (Figure 5 and 6).
We are grateful to have your helpful comments and for the precious time spent reviewing our article. According to your suggestions, we have additionally put the figure 2 and changed figure 5 and 6 with patient photography. We are confident that readers would be interested in this revised article.
Reviewer 3 Report
Congratulations on a very interesting study: "Anatomical guidelines for botulinum neurotoxin injection for glabellar frown lines" and excellent figures. Below I present a few comments that, in my opinion, will help to refine the manuscript.
Introduction:
- Sentence "Various studies relating BoNT injection points to specific muscle anatomy have already been published [10-20]." has too many references that aren't about the face.
- Refer in the introduction to the current systematic reviews of BoNT administration in the facial muscles, e.g. https://www.mdpi.com/2072-6651/13/2/169 https://www.mdpi.com/1660-4601 / 18/18/9552.
Anatomy and Injection Techniques:
- It is very important that you indicate whose research the knowledge you share comes from. This is completely missing in these paragraphs. Please add the relevant references.
- Is there any way to find out if oblique bellies are narrow / wide before giving BoNT? If so, please describe it.
- I conclude that the puncture point closer to the midline of the body was designed to be correct whether the belly is wide or narrow. I cannot find this information in the text. If it is so, and you have not written about it, I think it is worth emphasizing.
- You locate the injection sites based on the distance in centimeters. Are these distances not dependent on the features of the face, the distance between anatomical points, e.g. the width between the pupils? Perhaps it would be more appropriate to give the injection sites as a fraction of the distance between the anatomical points, so as to be independent of the size of the face.
Discussion:
- Consider creating a header for your conclusion paragraph. It would be clearer for me, but of course this aspect of the layout of the text is an individual matter and depends entirely on your judgment.
Once again, congratulations on your promising manuscript. Due to the shortage of references in the key chapters of Anatomy and Injection Techniques, I have to ask for a major revision.
Author Response
The authors are very grateful to the reviewer for their meticulous reading of the manuscript. The authors would like to kindly acknowledge the reviews. Here follow our answers to specific points of yours.
Congratulations on a very interesting study: "Anatomical guidelines for botulinum neurotoxin injection for glabellar frown lines" and excellent figures. Below I present a few comments that, in my opinion, will help to refine the manuscript.
Introduction:
- Sentence "Various studies relating BoNT injection points to specific muscle anatomy have already been published [10-20]." has too many references that aren't about the face.
- Refer in the introduction to the current systematic reviews of BoNT administration in the facial muscles, e.g. https://www.mdpi.com/2072-6651/13/2/169 https://www.mdpi.com/1660-4601 / 18/18/9552.
Answer: We have deleted the references and referred the reference the reviewer have suggested. “https://www.mdpi.com/2072-6651/13/2/169”
Anatomy and Injection Techniques:
- It is very important that you indicate whose research the knowledge you share comes from. This is completely missing in these paragraphs. Please add the relevant references.
- Is there any way to find out if oblique bellies are narrow / wide before giving BoNT? If so, please describe it.
Answer: We have cited the relevant references for the anatomy and injection technique paragraphs. We are sorry but, there is no way to find out oblique belies are narrow or wide before giving BoNT.
- I conclude that the puncture point closer to the midline of the body was designed to be correct whether the belly is wide or narrow. I cannot find this information in the text. If it is so, and you have not written about it, I think it is worth emphasizing.
Answer: We have cited the relevant references for the anatomy and injection technique paragraphs. There is no way to distinguish type of belies (narrow or wide). Thereby, we have stated “However, two type of the CSM muscle is difficult to distinguish and injection points for two types are exactly same.”
- You locate the injection sites based on the distance in centimeters. Are these distances not dependent on the features of the face, the distance between anatomical points, e.g. the width between the pupils? Perhaps it would be more appropriate to give the injection sites as a fraction of the distance between the anatomical points, so as to be independent of the size of the face.
Answer: As the reviewers suggestion we have set up anatomical landmarks that can be easily noticeable for the practitioner. The medial point should be interciliary point located on infraorbital notch that can be palpated and the lateral point lateral point was crossing point of mid-pupillary line and superciliary arch as the suggestion. Thank you for suggesting great idea!
Discussion:
- Consider creating a header for your conclusion paragraph. It would be clearer for me, but of course this aspect of the layout of the text is an individual matter and depends entirely on your judgment.
Answer: We have created header for the conclusion paragraph as suggested. Also, a paragraph of limitation of the study has been added.
Once again, congratulations on your promising manuscript. Due to the shortage of references in the key chapters of Anatomy and Injection Techniques, I have to ask for a major revision.
We are grateful to have your helpful comments and for the precious time spent reviewing our article. According to reviewers suggestions, we have additionally put the figure 2 and changed figure 5 and 6 with patient photography. We are confident that readers would be interested in this revised article.
Round 2
Reviewer 1 Report
- The statement regarding the mechanism of action of BoNT as “permanent blockage of…” remains incorrect.
- Please adjust all figure numbers accordingly after adding Figure 2.
- In the paragraph “A total of 6U…(Figure 4),” please clarify whether the location of the medial point as “infraorbital notch” is correct. This was also mentioned in the conclusion. “Infraorbital” was also used in the paragraph “BoNT was injected into…should point upward,” please clarify the accuracy of the terminology. I assume the authors meant the inferior border of the supraorbital rim, but not sure.
- Thanks for adding the reference! However, I could not find relevant evidence regarding the statement “BoNT spreads approximately 1 cm” in the added reference. Could you please verify to me or the editor(s) which sentence(s) you were referring to?
- Again, there is inconsistent use of “corrugator supercilii” and “corrugator supercilia.”
Thanks!
Author Response
Dear reviewer 1,
We are really thankful to have you as reviewer. There have been many critical errors. We would like to thank you for catching many errors that we should have caught. Your comment and points were really helpful and important to us.
Comment 1: The statement regarding the mechanism of action of BoNT as “permanent blockage of…” remains incorrect.
Answer: We certainly agree with your point. We have changed the statement as following "The botulinum neurotoxin (BoNT) mechanism of action is by binding presynaptically to high-affinity protein receptor and polysialogangliosides on the cholinergic nerve terminals and then enters the neuron and causes a sustained inhibition of synaptic transmission." The authors have referred “Botulinum neurotoxins: mechanism of action” (2013) by Ann P Tighe.
Comment 2: Please adjust all figure numbers accordingly after adding Figure 2.
Answer: We have adjusted all figure numbers as reviewer have pointed out. Thank you again for catching our mistake.
Comment 3: In the paragraph “A total of 6U…(Figure 4),” please clarify whether the location of the medial point as “infraorbital notch” is correct. This was also mentioned in the conclusion. “Infraorbital” was also used in the paragraph “BoNT was injected into…should point upward,” please clarify the accuracy of the terminology. I assume the authors meant the inferior border of the supraorbital rim, but not sure.
Answer: This was really important point and issue. Thank you! And we are sorry for the wrong naming of the anatomical point. The anthropology of the facial landmarks have been discussed in “Quantitative anatomical analysis of facial expression using a 3D motion capture system: Application to cosmetic surgery and facial recognition technology” (Clinical anatomy, 2015), the medial point we have mentioned should be “interciliary point located on frontal notch”.
Comment 4: Thanks for adding the reference! However, I could not find relevant evidence regarding the statement “BoNT spreads approximately 1 cm” in the added reference. Could you please verify to me or the editor(s) which sentence(s) you were referring to?
Answer: We are so thankful to have your comment for catching critical errors. We are sorry for the errors in cm of diffusion. We should carefully state as following, it should have been 1.5 to 3 cm in 1 U of botulinum neurotoxin. We have cited another reference to it. “Histologic Assessment Muscle Fiber Response of Dose-Related Diffusion and After Therapeutic Botulinum A Toxin Injections” by Borodic et al.
Comment 5: Again, there is inconsistent use of “corrugator supercilii” and “corrugator supercilia.”
We have corrected term to corrugator supercilii all over the manuscript. This time we have double checked. We are sorry for keep making spelling errors.
Overall, we as the authors regret and ashamed to have our shortness of attention to the first round of revised manuscript. Further, we will be more careful in revising the manuscript.
Reviewer 2 Report
All concerns have been addressed.
Author Response
The authors are very grateful to the reviewers for their meticulous reading of the paper. The authors would like to kindly acknowledge the reviews.
Reviewer 3 Report
Thank you for addressing all my suggestions. I have no more comments. Congratulations on an interesting article that I believe will be published.
Author Response

(The authors gave the same response as above.)

Round 3
Reviewer 1 Report
Thank you for addressing all my questions! I believe the article provides accurate information and is of clinical utility. Nice job on the illustrative figures. Minor revision may be required as follows.
“The botulinum neurotoxin (BoNT) mechanism of action is by binding presynaptically to high-affinity protein receptor and polysialogangliosides on the cholinergic nerve terminals, then entering the neuron and causing a sustained inhibition of synaptic transmission.”
“One unit of BoNT is reported to spread over 1.5 to 3 cm;”
Line 67, “Figure 2” should be “Figure 3.”
Line 76, “Figure 3” should be “Figure 4.”
Line 119-120, “Figure 4” should be “Figure 5.”
Line 139, “Figure 5” should be “Figure 6.”
Line 159, “Figure 6” should be “Figure 7.”
Line 208, “Figure 4” should be “Figure 5.”
Thanks!
Author Response
Dear reviewer,
The authors are very grateful to the reviewers for their meticulous reading of the paper. The requested changes have been made and high-lighted. The authors would like to kindly acknowledge the reviewer's effort for finalizing the manuscript.
Thank you again,
Sincerely,
Corresponding author